# Seasonal Trophic Ecology and Diet Shift in the Common Sole *Solea solea* in the Central Adriatic Sea

**DOI:** 10.3390/ani12233369

**Published:** 2022-11-30

**Authors:** Emanuela Fanelli, Elena Principato, Eleonora Monfardini, Zaira Da Ros, Giuseppe Scarcella, Alberto Santojanni, Sabrina Colella

**Affiliations:** 1Department of Life and Environmental Sciences, Polytechnic University of Marche, Via Brecce Bianche, 60131 Ancona, Italy; 2Stazione Zoologica di Napoli Anton Dohrn, Villa Comunale, 80100 Naples, Italy; 3National Research Council-Institute of Marine Biological Resources and Biotechnologies (CNR-IRBIM), Largo Fiera della Pesca 2, 60125 Ancona, Italy

**Keywords:** diet composition, feeding ecology, mesopredator, stable isotopes, stomach content analysis

## Abstract

**Simple Summary:**

The common sole is one of the most important commercial fish species landed in Europe. In this study, we analysed the trophic ecology of this species at the seasonal scale, coupled with data on its condition (by hepatosomatic and gonadosomatic indices). We observed significant seasonal changes in diet, clearly linked to the reproductive cycle only for females. Both the analysis based on stomach contents and on stable isotopes highlighted a greater contribution of prey of high energetic content before the spawning period, notwithstanding the opportunistic behaviour of this benthophagous species.

**Abstract:**

The common sole, *Solea solea*, is one the most important commercial species in Europe and, within the Mediterranean, the Adriatic basin is the most crucial area for its production. Although the species is overexploited in the basin, data on its trophic ecology are fragmentary, even though this is one of the most important features within the Ecosystem Approach to Fishery. Here, we analysed temporal variations in the feeding ecology of the species by using an integrated approach of stomach contents and stable isotope analyses coupled with the analysis of some condition indices such as the gonadosomatic and the hepatosomatic indices. Changes in diet and trophic level across the years in adult females were clearly linked to the different energetic requirements facing reproduction. Temporal changes throughout the year were mainly related to changes in food availability. This study confirms the opportunistic behaviour of this benthophagous species and its role as a mesopredator, opening new perspectives for further investigations on the effects of the overexploitation of this important fishery resource on the marine trophic web.

## 1. Introduction

The common sole*, Solea solea* (Linnaeus, 1758) is one of the most important fish species landed in Europe, having a high commercial value both for fisheries and aquaculture [1,2]. This species has a wide distribution as it is found both in the eastern Atlantic, from Scandinavia to Senegal, and throughout the Mediterranean, including the Gulf of Lion, the Ligurian Sea, the Ionian Sea, the Tyrrhenian Sea, the Aegean Sea, and the Adriatic Sea, while it is rare in the Black Sea [3]. The common sole is a demersal and sedentary species, living on sandy and muddy bottoms [4,5], the biology of which has been extensively studied, allowing to determine reproductive traits [4] and growth rates [6,7,8,9,10,11,12,13,14]. Briefly, in the Mediterranean Sea, the reproduction of common sole occurs from December to May [4,15] in the Adriatic from November to March [16] or from autumn to early winter [17]. The length at maturity is 25 cm [4,17,18]. The age of the first maturity is from 3 to 5 years [4]. Refs. [4,5] reported longevity of up to 24 years in females and 27 years in males.

The species is widespread in the Adriatic Sea, especially in the northern part. In the northern and central Adriatic, the distribution of this species depends on age: mature fish occur on the outer side of the Istrian coast, and younger fish (12–15 cm) are found in the Italian coastal waters, primarily at the mouth of the Po River [16]. Data from tagging experiments [19,20] showed that the majority of the Adriatic population was moving from north to south along the Italian coast and, probably, from south to north along the eastern Adriatic coast.

Information on the trophic ecology of the species mostly comes from the Atlantic coasts of northern Europe up to Portugal [21,22,23] and from the western Mediterranean [24,25]. These studies indicated the common sole as a benthic feeder, mainly preying on polychaetes, bivalves, and crustaceans. Notwithstanding the common sole is one of the most fished species throughout the northern Adriatic Sea (Mediterranean Sea, FAO subdivision GSA 17) [26], representing 23% of the landings of FAO-GFCM Area 37 (Mediterranean and Black Sea) [27], only a few studies analysed its trophic ecology in this area [28,29] during recent decades. This aspect is of particular importance in this basin considering the high fishing pressure on the species, the production of which is of about 2000 tons/year in the last decade in Italy [28]. In total, 64% of this biomass is caught using the “rapido” fishing gear [29], a typical bottom-trawl net used exclusively in this area. 

Studies of the diet of a benthic feeder species can be used as a useful indicator of changes in benthic communities [30,31]. Moreover, such studies if conducted over a large temporal scale can also provide insights on changes in food availability through time, which can be linked both to natural variations in benthic communities, changes in fishery pressure throughout the year, but also to the different feeding requirements of the species, linked to its reproductive traits. Seasonal reproductive processes respond to natural fluctuations in environmental factors [32]. The relationship between biological cycles and cyclic food availability has been established for different slope-dwelling species [33]. A relationship between the biological cycle and changes in feeding habits has been observed for other flatfishes [34,35], showing increased foraging during gonad development in pre-reproductive periods. 

The main objective of this study is to determine the trophic ecology of *Solea solea* using an integrated approach of both stomach contents (SCA) and of signatures of stable isotopes (*δ*^15^N and *δ*^13^C) (SIA) on specimens taken monthly over one year in the coastal fishing grounds off Ancona. SCA information is vulnerable to bias related to the material that is identifiable in the stomach and it only provides a snapshot of the diet [36]; SIA supplies information on the actually assimilated food [37]. In fact, different prey types are digested and evacuated at different rates, and the importance of rapidly digested prey may be underestimated [38,39]. Thus, the assessment of *δ*^15^N and *δ*^13^Cvalues, has become a standard technique in food web investigations [39]. The study of stable isotopes is widely applied to marine ecosystems [40] and when used in combination with stomach content analysis, it is a useful tool for exploring trophic interactions and dynamics in sympatric species [31,41,42,43]. Since the Adriatic Sea is the basin with the highest level of productivity for this flatfish species, the use of this combined approach may provide a comprehensive knowledge of the trophodynamics of such an important species for the benthic food web of the Adriatic basin. In this context, the specific aims of this paper are: (i) to assess seasonal changes in the feeding ecology of common sole in the central Adriatic Sea, and (ii) to determine the coupling between trophic dynamics and reproductive cycles.

## 2. Materials and Methods

### 2.1. Study Area

The Adriatic Sea is a semi-enclosed basin in the Mediterranean Sea [5]. It is conventionally divided into three sub-basins: North Adriatic, Central Adriatic, and South Adriatic. The sedimentology of the basin is determined by the regime of marine currents typical of the area and by fluvial inputs [44], specifically by the Po River flow. The Adriatic Sea provides ca. 30% of Mediterranean freshwaters (~30% of that comes from the Po River), determining a positive water balance of 90–150 km^3^ exported to the Mediterranean. The average salinity is 38‰, but in the northern area, this is lower and more variable, mostly depending on the Po River discharge. Water mass circulation is mainly driven by dominant winds (Bora and Scirocco) causing cyclonic circulation. Three different water masses dominate the Adriatic circulation: the Adriatic Surface Waters (AdSW), the Levantine Intermediate Waters (LIW), and the Adriatic Deep Waters (AdDW), which branches out into the Northern (NAdDW), Middle (MAdDW), and Southern (SAdDW) Adriatic Deep Waters. The area of interest for this study is GSA 17, i.e., the Northern Adriatic Sea (Figure 1). 

### 2.2. Samples Collection and Processing

Samplings were carried out in 2019, from January to December, in relation to planned research activities requested by the “Data Collection Framework-Biological Sampling of Commercial Catches”, conducted by the CNR-IRBIM (Institute for Marine Biological Resources and Biotechnology of the National Research Council) of Ancona.

Samplings occurred each month, excluding August that in the Northern and Central Adriatic coincided with the seasonal trawl fishing ban (M.D n. 173/2019). Samples were collected monthly from landings of a commercial *rapido* trawler registered in the Ancona port and operating in the fishing ground off Ancona (Central Adriatic Sea). The *rapido* gear is a modified beam trawl with a rigid mouth rigged with 5–7 iron teeth along the lower leading edge and a net bag to collect the catches [45]. According to Italian regulation, a square mesh of 40 mm is used (Council Regulation 1967/2006). The local fleet moves inshore–offshore according to the season, the weather, and the known migratory patterns of the targeted species. In the case of common sole, it is known that the species move inshore during spring close to the reproduction period [46]; thus, the sampling locations, being based on Fishery Dependent Data, reflect all these aspects. 

Once captured, the total length (TL, in mm) of all specimens was measured. The length frequency distribution (cm of Total Length, TL) was determined by measuring all specimens for each sampling day. The samples were divided into two groups according to the bimodal pattern observed in the TL data. Specimens with a TL < 26 cm were classified as “medium-size individuals”, while specimens with a TL > 26 cm were listed as “large-size individuals” (see below).

For each selected specimen, wet weight (WW, in g), sex, and maturity stage (according to the ICES protocol 2007) were recorded. The gonads and liver were also weighed, while the stomach was stored in sterile jars filled with 70% alcohol. Then, a portion of white muscle on the right side of the sole and close to the dorsal fin was dissected and stored at −20 °C in sterile test tubes for SIA. 

The gonadosomatic index (%GSI) is often considered a suitable index to evaluate the temporal variation of maturity stages of gonads in order to define the reproductive cycle [47,48] as a surrogate for reproductive effort, and it was calculated both for all the specimens and for female and male specimens separately as: %GSI = 100 × (gonad weight/body weight). The hepatosomatic index (%HSI), calculated as %HSI = 100 × (liver weight/body weight), was determined as a proxy of metabolic activity [33].

For each stomach, fullness (as (stomach content weight/the total body weight) × 100) was calculated as a proxy of feeding intensity. The stomach contents were analysed under a stereomicroscope (Leica Wild L3B) to classify the ingested organisms at the lowest taxonomical level as possible. An empirical evaluation of the status of prey digestion was also attributed to each prey, using value 1 for intact prey, 2 when the prey was partially digested, and 3 when it was highly digested [49]. The following feeding indices were then calculated [36]: percent frequency of occurrence (%F), percent numerical composition (%N), and percent of gravimetric composition (%W). Trophic diversity was calculated for each season based on the Shannon–Wiener H′ index [50].

### 2.3. Stable Isotopes Analysis

For SIA, samples taken from each selected specimen were oven-dried at 60 °C for 24 h, homogenized, and powdered with a grinder. A minimum of 0.7 mg (up to 1.2 mg) from each powdered sample was then weighed and put into tin capsules (Elemental Microanalysis Tin Capsules Pressed, Standard Weight 5 × 3.5 mm) [51].

Carbon and nitrogen contents were determined using an elemental analyser (ANCA-GLS 20–20 series) coupled to an isotope ratio—mass spectrometer (SERCON) according to standard protocols [30].

The *δ*^15^N and *δ*^13^C values were expressed in parts per thousand (‰) and were quantified relatively to Vienna Pee Dee Belemnite (VPDB) and atmospheric N_2_ standards, respectively, according to the following formula:*δ*^13^C or *δ*^15^N (‰) = [(R_sample_/R_standard_) − 1] × 103
where R = ^13^C/^12^C or ^15^N/^14^N.

The total mass percent proportions of C and N were used to calculate C:N ratios. Lipids were not extracted from the samples. A correction equation was applied to the δ^13^C values (when the respective C/N values were higher than 3 [52]) by using the relationship between the C:N ratios and the *δ*^13^C signatures according to [53]: *δ*^13^Ccorrected = *δ*^13^C_untreated_ − 3.32 + 0.99 × C:N_bulk_. If significant differences occurred between untreated *δ*^13^C and corrected *δ*^13^C (according to Student’s t-test), corrected *δ*^13^C was used for the following analyses.

### 2.4. Data Treatment

#### 2.4.1. Biological Indices, Stomach Contents, and Trophic Diversity

Seasonal values of biological indices (%GSI, %HSI and fullness) and H’ were tested for differences by using univariate analyses based on a crossed design with two fixed factors: “Season”, with four levels (spring: March–May, summer: June–July, Autumn: September–November, and winter: December–February) and “Size” (fixed with two levels: medium-size, M, from 19–25 cm, and large-size, L, specimens from 26–35 cm). A PERMANOVA main test was conducted on the Euclidean resemblance matrix of untransformed data [54]. When significant differences were encountered, pair-wise comparisons were carried out to identify the source of variation. Significance was set at *p* < 0.05; *p*-values were obtained using 9999 permutations of residuals under a reduced model [54] from Monte Carlo asymptotic distributions.

Changes in the diet composition of the common sole across seasons and between sizes were tested through multivariate analyses of the SCA results. First, an nMDS (non-metric Multi-Dimensional Scaling) [55] was run on the Modified Gower resemblance matrix of the 4th-root transformed prey-biomass data to visualize the seasonal pattern in the diet. The 4th-root transformation allows the reduction in the excessive contribution of highly abundant species [54,56], and the “Modified Gower” distance [57] is considered the most suitable measure for stomach content data, which generally contain many zeros [54]. Then, a PERMANOVA test was carried out on the same resemblance matrix, to investigate temporal and size-related changes in the species’ diet.

A SIMPER analysis [56] was applied to 4th-root transformed prey-biomass data to determine the most typical prey species in each season and for medium- and large-size specimens.

Finally, two Generalized Linear Models—which are flexible generalisations of ordinary least squares regression—were run with the fullness of females and the fullness of males separately, as dependent variables, vs. total length, weight, maturity, %HSI, and %GSI. The distribution family used was Gaussian. The model selection was based on minimising Akaike’s Information Criterion (AIC) values.

All analyses were performed using PRIMER6 and PERMANOVA+ [54,56] and R [58].

#### 2.4.2. Stable Isotopes Data

Univariate and multivariate analyses were conducted on the SIA results obtained for large-size individuals of *S. solea*. The PERMANOVA test was carried out using PRIMER6 and PERMANOVA+ [54,56] on Euclidean distance resemblance matrices of untransformed *δ*^13^C and *δ*^15^N values separately and of these two variables selected together to detect significant differences among the factors “Season”, “Size”, and their interaction. Correlation among TL and, separately, variations in the *δ*^13^C and *δ*^15^N values were tested using R [58] with the function *cor.test*.

Simmr (Stable Isotope Mixing Models in R) R-package [59] was used to provide an estimate of the relative contributions of different food sources to the isotopic content of *S. solea*. This package is designed to solve mixing equations for stable isotopic data within a Bayesian framework and is an upgrade of the package SIAR (Stable Isotope Analysis in R) [59]. The simmr models were fitted with a Trophic Enrichment Factor (TEF) of 0.9 ± 0.1‰ for ^13^C and a TEF of 2.7 ± 0.1‰ for ^15^N, according to [60,61]. 

Before running the simmr, the best mixing polygon was determined in order to identify the main sources to be used in the model itself [62,63]. The main food items identified with SCA were considered as potential food sources. We used the mean isotopic contents of pelagic fishes (*Sardina pilchardus* and *Engraulis encrausicolus* from Fanelli et al., 2022 [64]) and ichthyoplankton (hereafter, “Fish larvae”) taken from [65].

Many polychaetes were found with SCA. The isotopic values of a common Adriatic species, *Sternaspis scutata*, were taken from [66]. The values of the isotopic contents of polychaetes species belonging to the family of Nephtyidae living in the Adriatic Sea were taken from the literature [67]. Several specimens of crustaceans belonging to the orders Amphipoda and Tanaidacea were found with SCA. *Ampelisca* sp. and *Apseudopsis latreillii* (both items found in stomachs contents) were used as potential food sources for determining the best mixing polygon, taking their isotopic contents from specimens living in the Adriatic Sea, in an area near that in which samplings were conducted [67]. For the phylum of Echinodermata, we considered the class of Holothuroidea using the mean isotopic contents of two Adriatic common species of the family of Cucumariidae whose spiculae were found in the stomach contents (*Ocnus planci* and *Paraleptopentacta elongata*) (values from Fanelli E., unpublished data). For Mollusca, we considered isotope values obtained from the analysis of samples of *Anadara demiri* and *A. kagoshimensis*, *Mimachlamys varia, Chamalea gallina*, *Donax* spp., *Spisula* sp., *Tellina* spp., and *Nassarius* sp., all collected in the same sampling sites (values from Fanelli E., unpublished data). The SIBER package (Stable Isotope Bayesian Ellipses in R) [68] was used to determine the isotopic niche width of each season and the trophic preferences of *S. solea*. We considered one community with 4 groups, each one composed of the isotopic contents of the specimens collected in the four different seasons. To underline the differences between sexes, we run the SIBER model considering male and female specimens as two separate communities. In both cases, SIBER was used to calculate the total area of the convex hull (TA), the corrected Standard Ellipse Areas (SEAc), and the mean distance to the centroid (CD) [69]. The TA gives an indication of the variety of food sources on which the species can feed, while SEA_C_ (which contains approximately 40% of the data within a set of bivariate data) represents the core area for a population or community [68,69]. CD estimates trophic diversity within a food web and is a function of the degree of group spacing. The command maxLikOverlap of the SIBER package was used to calculate the overlap between consecutive pairs of SEA_C_s.

## 3. Results

### 3.1. Population Structure

A total of 477 specimens of *S. solea* were captured during the sampling carried out in 2019. In total, 138 individuals were collected in spring, 73 in summer, 120 in autumn, and 146 in winter. The percentage of female individuals ranged from 59 to 70% in spring and winter, respectively (Appendix A). The percentage of males ranged from 30 to 41% in winter and spring, respectively (Appendix A). 

Considering all the specimens, the mean observed TL was 26.6 ± 3.4 cm (mean TL of 25.6 ± 3.5 cm for females and 22.6 ± 2 cm for males). The length frequency distribution (Figure 2) shows that most of the males had a length comprising between 19 and 25 cm, while individuals with higher TL were mainly females (Figure 2). The bimodal curve identified in Figure 2 allowed the division of the collected population into medium-size (M) and large-size (L) individuals with a threshold of 26 cm of TL.

### 3.2. Variations in Condition Indices and Fullness

The %GSI varied between 0.38 ± 0.31 in medium-size individuals in summer and 5.29 ± 3.12 in large individuals in winter (Figure 3A and Appendix A). PERMANOVA analysis showed that %GSI varied significantly among the seasons and between the two different sizes (Appendix A). The pair-wise comparison for the factor “Season” within level “M” of factor “Size” showed differences between the %GSI recorded in winter and spring (*p* ≤ 0.01), while the %GSI of large-size individuals varied between summer and autumn (*p* ≤ 0.001), and winter and spring (*p* ≤ 0.01).

The %HSI varied between 0.85 ± 0.23 in medium-size individuals in summer and 1.52 ± 0.47 in large-size individuals in winter (Figure 3B and Appendix A). PERMANOVA analysis showed that %HSI varied significantly among the seasons and between the two different sizes (Appendix A). The pair-wise comparison for the factor “Season” within level “M” of factor “Size” showed significant differences between the values of %HSI observed in spring/summer and in autumn/winter (*p* ≤ 0.001 in both cases). The pair-wise comparison for the factor “Season” within level “L” of factor “Size” did not show significant differences in the values of %HSI.

Stomach fullness (%) in females ranged between 0.12% in medium-size individuals in spring and 0.29% in large-size individuals in summer (Figure 3C and Appendix A), but the PERMANOVA main test did not highlight significant differences (Appendix A). Additionally, the PERMANOVA main test conducted on the values of stomach fullness measured separately in females and males of *S. solea* did not show significant seasonal variations or significant differences between the two sizes (Appendix A). The values of %GSI, %HSI, and stomach fullness in females (Figure 3D) were highest in winter and lowest in summer. Differences in the values of %GSI measured in females showed significant differences for both factors, “Season” and “Size” (*p* ≤ 0.001). For medium-size females, the %GSI was significantly higher in winter than in spring (*p* ≤ 0.5). For large-size females, the %GSI was significantly higher in winter than in spring (*p* ≤ 0.5) and also significantly higher in autumn than in summer (*p* ≤ 0.01). Differences in the values of %HSI measured in females showed significant differences for both factors “Season” and “Size” (*p* ≤ 0.001 and *p* ≤ 0.01, respectively). For medium-size females, the %HSI was significantly higher in spring than in summer and also in winter than in autumn (*p* ≤ 0.001 in both cases). For large-size females, no seasonal differences were found in values of %HSI.

According to the GLM model results, the %GSI and %HSI were the foremost explanatory variables of the fullness of females of *S. solea* throughout the sampling period, with positive and negative variations with %HSI and %GSI, respectively (Table 1). The AIC for the female model was 82.15. Regarding males, any of the tested explanatory variables resulted in being significant to explain changes in fullness.

### 3.3. Seasonal Variations in Diet Composition

The results obtained with SCA (Figure 4 and Appendix A) showed that, along the whole year, the diet of *S. solea* was mainly composed, in terms of %W, by Polychaeta (mean %W of 27%) and Mollusca (mean %W of 20%), followed by Crustacea (mean %W of 15%) (Figure 4A). Considering the overall results, 33% of the stomach contents included parasites, sediment particles, fish scales, fish eggs, and vegetal remains (considered together as “other material”). Several fish fragments and fish scales were identified with SCA, with higher values of %W in winter (Appendix A). Fish larvae were found in the stomach contents mainly in specimens collected in spring (Appendix A). Several parasites belonging to the class of Cestoda were found in *S. solea* stomachs, with a higher %W in summer and autumn than in the rest of the year (Appendix A).

The PERMANOVA main test carried out on the diet composition of *S. solea* within the year (%W) highlighted significant differences (*p* ≤ 0.001 for both factors, Appendix A). Larger animals showed significant differences in the diets of autumn/winter and of winter/spring (*p* ≤ 0.001 and *p* ≤ 0.5), while medium-size individuals had variable diets among all the pairs of consecutive seasons (Appendix A). Within size classes, the diets of both medium and large soles differed between spring and winter (PERMANOVA main test on %W, *p* ≤ 0.001 in both cases). These differences are explained by SIMPER results that show how the contribution of Polychaeta and fish larvae in spring and the presence of fish scales together with the contribution of fish larvae and Polychaeta in winter caused these variations (Figure 4 and Appendix A)

The SIMPER results confirmed that the main contribution to the dissimilarity between the medium- and large-size individuals was given by Polychaeta/fish larvae and scales in spring, Polychaeta/shells/fish larvae and *Ampelisca* sp. in summer, Polychaeta/*Ampelisca* sp., fish scales and Mollusca in autumn, and fish scales, larvae/Polychaeta/fish eggs, and *Ampelisca* sp. in winter (Appendix A).

SIMPER also confirmed that the diet composition of *S. solea* in spring, summer, and autumn was dominated by polychaetes and fish larvae, while in winter, the *S. solea* mainly fed on fish (Appendix A).

Trophic diversity (H’) varied among seasons (Appendix A), with higher values in spring than in summer (*p* > 0.05) and higher values in winter than in spring (*p* > 0.001). Trophic diversity was higher in the medium-size soles (*p* < 0.001).

### 3.4. Seasonal Variations in Stable Isotope Composition 

A total of 68 samples of muscles taken from medium- and large-size specimens of *S. solea* were analysed to determine the seasonal content of stable isotopes of C and N. The values of *δ*^13^C varied between −18.4 ± 0.5‰ in spring and −17.7 ± 0.5‰ in summer and winter for medium-size individuals and between −17.7 ± 0.8 ‰ in winter and −17.3 ± 0.5‰ in spring for large-size individuals (Figure 5A and Appendix A). The values of *δ*^15^N varied between 11 ± 0.6‰ in summer and 11.9 ± 0.5‰ in winter for medium-size individuals and between 11.2 ± 0.2‰ in spring and 12 ± 0.6‰ in winter for large-size individuals (Figure 5B and Appendix A). PERMANOVA main test showed significant differences (*p* ≤ 0.05) in *δ*^13^C-*δ*^15^N values for factors “Season” and “Size” (Appendix A). The *δ*^15^N values differed by “Season”, while those of *δ*^13^C varied significantly between the two levels of the factor “Size”. The values of C:N ranged from 2.12 in large-size individuals captured in winter to 2.39 in medium-size individuals captured in spring. The C/N values did not vary significantly for any tested factors. The mean seasonal *δ*^15^N values were significantly higher in winter than in autumn and spring (*p* ≤ 0.001 and *p* ≤ 0.05).

No correlation was found between the *δ*^13^C values and TL (*p* > 0.05) or the *δ*^15^N values and TL (*p* > 0.05). Considering the lack of a correlation between isotopic values and TL, simmr and SIBER were run by combining size classes and running simmr and SIBER on one group for each of the four seasons.

### 3.5. Bayesian Mixing Models and SEAc

Since no correlation between *δ*^13^C values and *δ*^15^N values and TL were found, Bayesian mixing models were run on all the isotopic results without distinguishing between medium- and large-size specimens (Appendix A). The family of Cucumariidae that includes *Ocnus planci* and *Paraleptopentacta elongata*, fish larvae, *Ampelisca* sp., the nepthyid polychaetes and the pelagic fish were found to be the sources that produced the best mixing polygon for *S. solea* (Figure 6). The isotopic values of Mollusca specimens fell inside the mixing polygon near the sole samples. Due to this, they were not used to build the best mixing polygon.

The output of the simmr model provided the proportional contribution of each food source to the diet of *S. solea* in each season (Figure 7 and Appendix A). In spring, summer, and winter, polychaetes gave the main contribution to the diet of *S. solea* (40, 38 and 4%, respectively), while in autumn, the main contribution was given by pelagic fishes (33%). Across the year, the mean proportional contributions of pelagic fishes ranged between 22 and 33% (in summer and autumn, respectively), those of Nephtyidae from 28 to 48% (in autumn and winter, respectively), and those of *Ampelisca* sp. varied from 7 to 24% (in winter and autumn, respectively). Minor contributions were given by fish larvae and holothurians of the family Cucumariidae. The mean proportional contributions of fish larvae ranged from 5 to 9% (in autumn/winter and in spring, respectively). The mean proportional contributions of Cucumariidae ranged from 10 to 18% (in autumn/winter and in spring, respectively).

SIBER outputs showed that *S. solea* has the widest SEA_C_ in spring and the narrowest SEA_C_ in autumn (Table 2 and Figure 8). The distance from the centroid (CD) was the highest in spring and the lowest in autumn. The overlap between the trophic niches of spring and summer was 80%, while that between the trophic niches of summer and autumn was 56% The autumn and winter trophic niche overlap was 8%, while that of the trophic niches of winter and spring was 33%. SIBER for males and females are reported in Appendix A.

## 4. Discussion

### 4.1. Seasonal Changes in the Feeding Ecology of Solea solea

Like many benthic organisms, *Solea solea* lives motionless and sunken on sandy and muddy bottoms during the day [4,15,17]. Its lifestyle, therefore, affects the diet and choice of prey. Flatfishes have asymmetrical jaws [70,71], a narrow stomach, and a long intestine [72,73]. Previous studies confirmed the strictly bentophagous-trophic habitus of the species. Indeed, according to studies on the trophic ecology of the species carried out in different areas of the Mediterranean, Northern Europe, and the North Atlantic, the common sole feeds primarily on benthic invertebrates such as polychaetes and crustaceans [4,17,22,24,25], and also molluscs [13,74,75]. Among crustaceans, amphipods prevail as chosen food items, in accordance with what has already been observed in the estuary of the Ebro River in Spain [24]. Many of the stomachs analysed were devoid of prey and food remains or were only partially full. This indicates a low rate of predatory activity of *S. solea* [72]. The high digestion rates are favoured by the lack of pyloric caeca, together with small dimensions of the oesophagus and stomach [23]. Furthermore, a high evacuation rate between the stomach and intestine confirms a lack of a storage phase of food in the stomach and a lack of pre-digestion [13]. Additionally, these features could justify the low values of stomach fullness found in our study. This kind of digestive apparatus also suggests that specimens of *S. solea* mainly feed on easily digestible prey such as polychaetes and sipunculids that live buried in the sediment. 

The highest values of trophic diversity were measured in winter, while the lowest values were recorded in autumn in agreement with various authors [70,71,76,77,78]. This is consistent with the seasonality of the dominant prey of the common sole. The dominance of these organisms in the sole diet is supported by other studies carried out in the northern hemisphere [70,71,73,76,77,78]. The diet rich in polychaetes also contains amphipods (constant throughout the year with lower values in spring), fish remains (especially in autumn), and molluscs (especially in winter) in accordance with findings from other areas [13,24,74,75,79,80,81]. The variability of the diet could indicate a greater availability or irregular distribution of the main resources for the species. The same observation has been reported for the Malabar sole, *Cynoglossus semifasciatus*, [73,82]. The presence of sediment in the stomach contents, recorded in other flatfish [73,77,82], is probably correlated to accidental ingestion during the process of searching for food.

Crustaceans were mainly found in the stomachs of common soles collected in summer and autumn. Probably during these two seasons, these prey are easier to capture due to their higher abundance above sediments. Thus, the probably accidental ingestion of sediment is lower because soles do not have to excavate under the sea bottom to capture prey. The SCA results highlighted the presence of unidentifiable fish remains and fish scales, classified with the label “other material”. Those items were abundant in the stomach contents of soles captured during autumn and winter, and fish remains were partially or highly digested. Due to their strictly benthic habits, it is unlikely that soles predate directly on pelagic fishes. More likely, they can be consumed after being discarded [62,83]. Bottom trawlers, pelagic trawlers, purse seiners, and small-scale fisheries together are responsible for around 60% of the total discarded biomasses [84]. Since the Northern-central Adriatic Sea is one of the most intensively fished areas, especially by trawlers [85], and since species like *Engraulis encrasicolus* and *Sardina pilchardus* are among the main species of pelagic fishes discarded in this basin [26], they may not naturally be a large part of the diets of common soles. This supports the hypothesis that common soles can feed opportunistically on bycatch, and this could also explain the disintegrated status of these prey since they are consumed above discard grounds [49,86].

The results showed that it is particularly useful to integrate multiple methods in trophic ecology, such as SCA and SIA [87]. In this case, the SIA results are consistent with the SCA results. The simmr outputs confirmed the predominance of polychaetes in the diet of common soles throughout the year, with the highest contribution in winter. Pelagic fishes or their remains were mainly assimilated during autumn, probably when the availability of other prey is lower. The higher percentage of crustaceans captured in summer and autumn resulted in a higher assimilation rate of these items (in particular, *Ampelisca* sp.) shown by the output of the Bayesian mixing model. An important contribution in spring and summer diets, that was not equally detected by SCA, was given by holothurians. Probably, the difference between the SCA and SIA results is caused by the difficult identification of holothurians’ spiculae in stomach contents and by the high digestion rates typical of soles that allowed to find only portions of these prey, hindering the estimation of their real ingestion amount. Conversely, simmr highlighted that, despite their high %W found in SCA (shells were not considered in this value but included in the voice “other material”), the molluscs we used in our simmr model were not among the main assimilated prey. Another issue to be considered in evaluating these biases between the obtained results is that diet is isotopically reflected in tissues a few months after ingestion [88]. Further investigations are needed to allow more inferences in this regard and to better determine the delay in the assimilation of the isotopic signals of some items by common soles, e.g., holothurians and molluscs.

### 4.2. Biological Traits of Common Sole in the Adriatic Sea and Correlation with Feeding Habits

As for the seasonal variation of the somatic indices, the %GSI shows higher values for both sexes in winter. This trend agrees with the data provided by the Data Collection Framework which made it possible to identify the reproductive period of the species from September to April [4,15,89]. As for the HSI index, there are very few bibliography studies that examine the Mediterranean basin and specifically the Adriatic sub-basin. In the case of females, the increase in this parameter in autumn and winter may be linked to the reproductive period. The peak in autumn is due to the fact that females invest a lot of energy in the subsequent deposition of the oocytes as described for other species [90,91,92,93].

The trend of fullness, which is a proxy of food intensity, is justified by the fact that females in winter and spring invest all of their energy in reproduction, as happens for other species [47,90,91,94]. In spring, the common sole has a wider trophic niche that allows specimens to focus more on reproduction than on feeding, while in winter, soles seemed to be more selective and chose to feed on abundant prey like polychaetes and discarded fishes, preserving energy for the subsequent reproduction in spring. Moreover, some prey items show an annual life cycle where edible sizes might be missing periodically, and this could influence the rate of ingestion.

No differences in the overall trophic niches of males and females were highlighted, but since we did not have many males among the samples analysed, further investigations are needed to determine differences between sexes.

In the isospace, the common sole shares its position with other taxa that show a *δ*^15^N content comprising between 9 and 12‰, which allows us to define them all as “marine mesopredators”, like *Raja asterias* [95], Scyliorhinidae, Triakidae [96], crabs (*Medorippe lanata*, *Goneplax rhomboides,* and *Liocarcinus* spp.), *Nephrops norvegicus* [83], cephalopods such as cuttlefish, and the carnivorous/scavenger gastropod *Nassarius* sp. Many of these are species of commercial interest, and understanding better their role in the trophic web will allow the more conscious management of fishery resources. The cascade effects of the removal of these mesopredators due to fishing efforts are not predictable and more attention of the scientific community has to be focused on this issue.

In winter and spring, large-size animals are focused on reproduction and their diets are based on a few very abundant prey, mainly molluscs and polychaetes. On the contrary, medium-size individuals can diversify their diets more in these two seasons. These outputs are not reflected in the SIA results, meaning that the species of molluscs we considered in building the best mixing polygon were likely not those ingested and assimilated by our targeted species, and further analysis is needed to determine better which molluscs contribute to soles’ diet.

The analysis of the data also underlines a greater presence of parasites in autumn. The presence of these parasites was reported in the intestine, but not in the stomach [97]. It is to be assumed that the abundance of these parasites, which have not been ingested voluntarily by the fish, increases in the reproductive period when the immune defences are low [98]. Finally, a great abundance of fish scales and other calcareous structures not identified, observed in the stomach contents, pointed to non-selective feeding. Common soles seemed to not eat whole fish, but fish remains found on the sea bottom. This abundance of fish remains on the sea floor probably indicates that they are discarded during intense fishing activity in the area [99,100]. In this study, the greatest abundance of fish remains was observed in the spring and winter months and could be attributed to the greater number of days at sea spent by fishermen.

## 5. Conclusions

The results of this work have contributed to the knowledge of the trophic ecology of the *Solea solea* species in the central Adriatic Sea. From the data obtained, it is possible to confirm that the high variability in the diet was related to both prey availability and energy requirements for reproduction. This study highlights that *Solea solea* is also an opportunistic species and a generalist benthivore, with a preference for more energetic prey before spawning. The availability of some of its prey is linked also to the fishing pressure as demonstrated by the abundant presence of fish remains, especially in the period following the summer season, where the days at sea, numerically speaking, are much more numerous and cause a larger amount of fishing discards, and soles can take advantage of these discards.

Further investigations on sexual differences and on the role of some prey (such as molluscs of commercial interest) in soles’ diet are needed in order to obtain more information for the management of this important resource considering also its role as a mesopredator in the Adriatic basin and the Mediterranean Sea.

## Figures and Tables

**Figure 1 animals-12-03369-f001:**
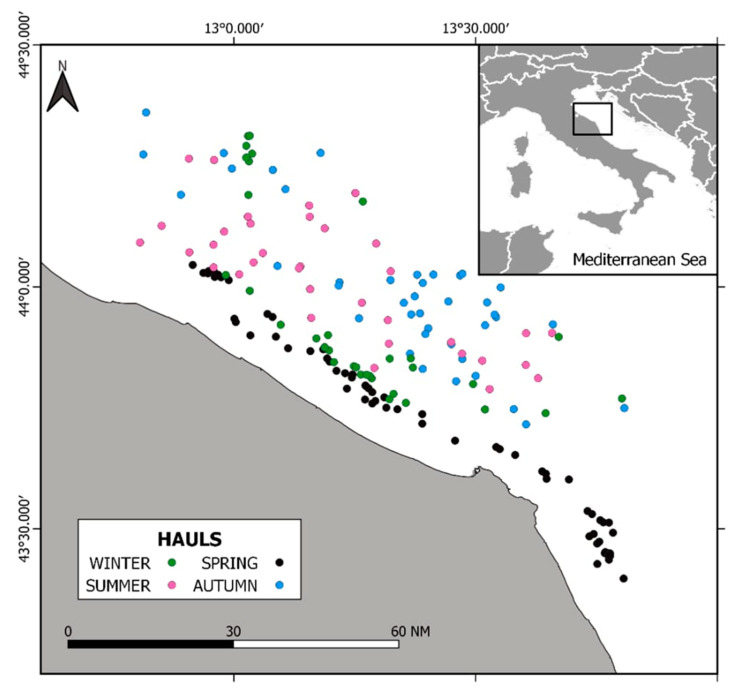
Sampling locations for a study of *Solea solea* diets in the Central Adriatic Sea in 2019.

**Figure 2 animals-12-03369-f002:**
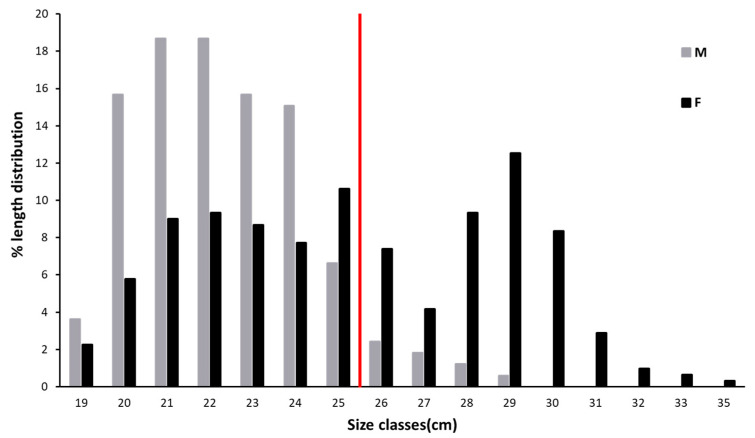
Length frequency distribution of *n* = 477 specimens of *Solea solea* collected from January to December 2019 in the Central Adriatic Sea for a study of their diet (the red bar separates specimens assigned to medium size 1, hereafter medium size (M) to those of size 2, hereafter large size (L) (threshold = 26 cm).

**Figure 3 animals-12-03369-f003:**
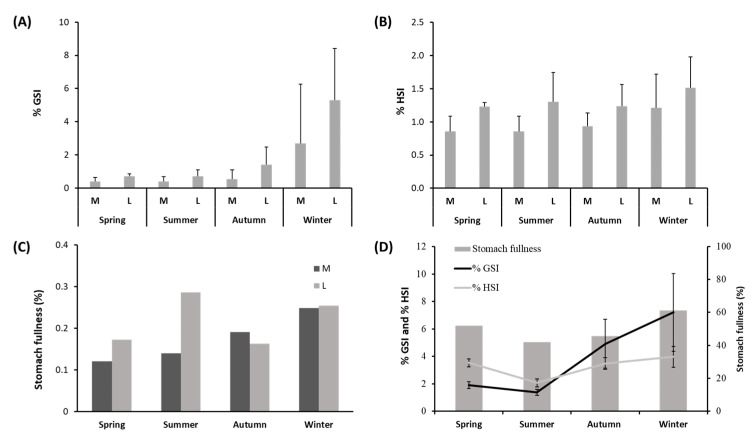
(**A**) %GSI, (**B**) %HSI, and (**C**) stomach fullness measured seasonally in medium–(M)- and large–(L)-size specimens of *Solea solea* captured from January to December 2019 in the Central Adriatic Sea for a study of their diet. (**D**) %GSI, %HSI, and stomach fullness measured seasonally in female specimens of *Solea solea*.

**Figure 4 animals-12-03369-f004:**
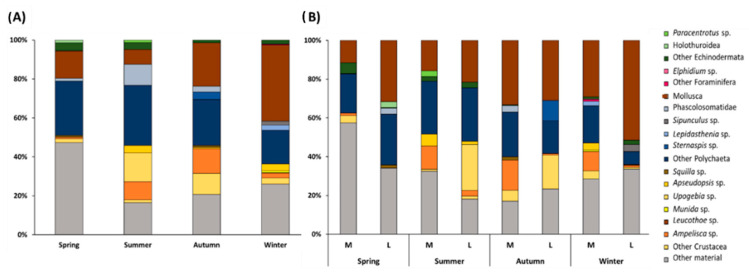
(**A**) Seasonal composition (%W) of the diet of *Solea solea* and (**B**) of the diet of medium- (M) and large-size (L) individuals captured from January to December 2019 in the Central Adriatic.

**Figure 5 animals-12-03369-f005:**
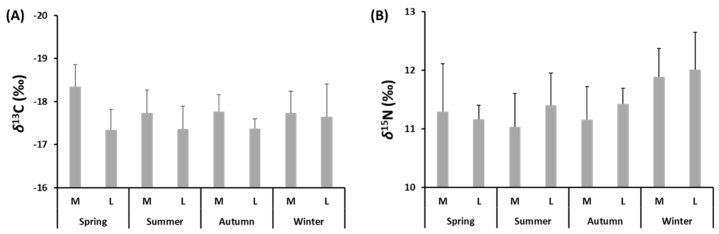
Values of (**A**) *δ*^13^C (‰) and (**B**) *δ*^15^N (‰) of medium (M)- and large (L)-size specimens of *S. solea* captured from January to December 2019 in the Central Adriatic Sea for a study of their diet.

**Figure 6 animals-12-03369-f006:**
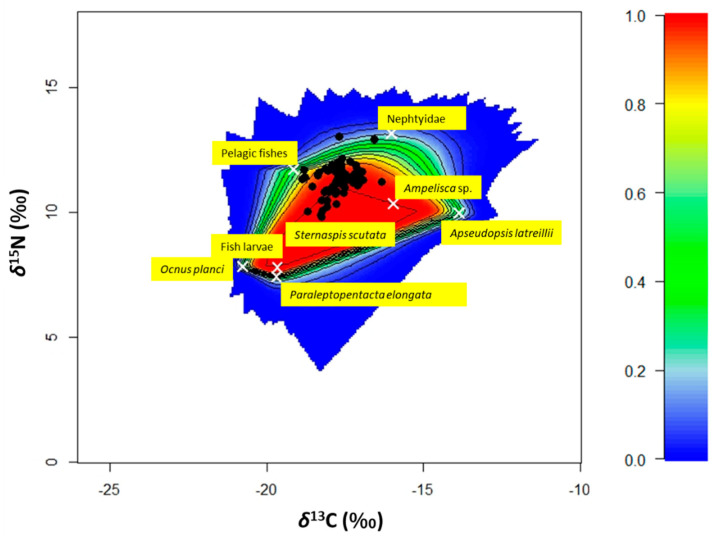
The biplot shows the simulated mixing with the positions of the specimens (*n* = 68) of *Solea solea* (black dots) captured in the Central Adriatic Sea between January and December 2019 and the average signatures of the sources selected for determining the best mixing polygon (white crosses). Probability contours are at the 5% level (outermost contour) and at every 10% level, following the colours shown in the legend in the right part of the graph.

**Figure 7 animals-12-03369-f007:**
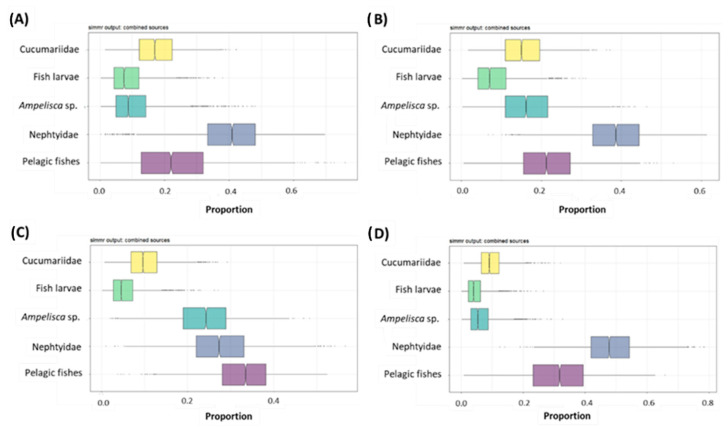
Posterior probabilities for the proportional contribution of each food source to the diet of *n* = 68 specimens of *Solea solea* captured from January to December 2019 in the Central Adriatic Sea in (**A**) spring, (**B**) summer, (**C**) autumn, and (**D**) winter obtained with stable isotope analysis mixing models. Each plot shows the proportions for each food. The horizontal line within the box represents the median and the 95% credibility interval, boxplots represent the 25th%, 50th%, and 75th% percentiles, while the whisker lengths represent the 1.5 interquartile range.

**Figure 8 animals-12-03369-f008:**
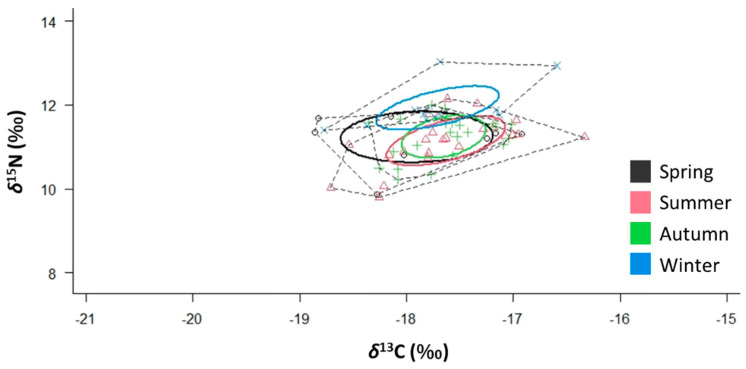
*δ*^13^C–*δ*^15^N scatterplot with seasonal standard ellipses corrected for a small sample size population (SEA_C_) overlaid for the specimens (*n* = 68) of *Solea solea* captured in the Central Adriatic Sea between January and December 2019 (*p* interval = 0.4).

**Table 1 animals-12-03369-t001:** Generalised linear models performed on fullness values of female *Solea solea* specimens captured from January to December 2019 in the Central Adriatic Sea for a study of their diet. +/−: sign of the correlation. HSI and GSI are hepatosomatic and gonadosomatic indices, respectively. AIC: Akaike’s information criterion.

	Df	Deviance	Resid. Df	Resid. Dev	F	Pr(>F)	
NULL			308	10.03			
HSI	1	0.55	307	9.48	17.92	0.00003	+
GSI	1	0.15	306	9.33	5.07	0.03	-

Null deviance: 24.08 on 312 degrees of freedom. Residual deviance: 23.23 on 310 degrees of freedom.

**Table 2 animals-12-03369-t002:** Values of the Total Area of the convex hull (TA), the corrected Standard Ellipse Area (SEA_C_), and the distance from the centroid (CD) calculated considering the factor “Season” on the isotopic contents of *n* = 68 specimens of *Solea solea* captured from January to December 2019 in the Central Adriatic Sea for a study of their diet. TA and SEA_C_ are expressed as ‰^2^.

Season	TA	SEA_C_	CD
Spring	2.1	1.5	0.83
Summer	2.8	0.9	0.66
Autumn	1.6	0.6	0.56
Winter	1.8	0.8	0.58

## Data Availability

Not applicable.

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
