# Peer review of "Seasonal Trophic Ecology and Diet Shift in the Common Sole Solea solea in the Central Adriatic Sea"

_animals, 2022, doi:10.3390/ani12233369_

Round 1

Reviewer 1 Report

This manuscript presents the results of a comprehensive study of the trophic ecology of Solea solea in the Central Adriatic Sea, including temporal variations among life stages and differences among females during reproductive season. Dietary analyses included stomach content analysis (SCI), stable isotope analyses for nitrogen and carbon (SIA), and assessing the relationship between tropic dynamics and reproductive cycles. All sampling and analytic protocols are well described, and appropriate statistical and modeling procedures are used.

The results of the study highlight several key factors contributing to the feeding behavior of S. solea, including the consumption of seasonally abundant opportunistic prey, and the energetic demands of reproduction in females. This is a valuable contribution to the understanding of the trophic ecology of an important resource species in the Adriatic Sea and should be broad interest to fishery biologists and management decisions.

Author Response

We really thank Reviewer 1 for their very positive comments that recognize our efforts in carrying out this study and the importance of the obtained results. We hope that addressing the comments of the other Reviewers and revising the manuscript also the English language and style are now improved.

Reviewer 2 Report

In my opinion this is a generally well-executed mensurative study of how common sole diets vary by season, sex, and size. The authors used a good combination of gut content analysis and stable isotope analyses (N and C) to determine which prey items were consumed and assimilated by the sole. N and C are commonly used, and this will allow the work to be compared across taxa. Sample sizes are large, so results should be robust.

There are a number of results that will interest fish biologists, and the study approach will be of general interest. For example, the authors found intriguing evidence that sole may be feeding largely on discarded, decomposing bycatch during the fishing season. It would be good to follow up on these results with DNA analyses of some of the fish samples that are currently listed as fish fragments, or morphological analysis if distinctive bones remain. Evidence of pelagic fishes would strengthen the bycatch idea. Although the manuscript often refers to 'pelagic fishes' I could find no evidence that fish material was actually identified.  Therefore their habitat cannot be established with the information provided. I'm not saying that this ms requires a DNA analysis, but the phrasing ought to be more cautious.

Also, the manuscript shows that marine worms are very common in sole diets.This is not unexpected, but it is good to see the gut content results bolstered by the stable isotope results.

In general analyses were appropriate and results were strong. However, the authors need to correct their P values for their many T tests for multiple comparisons, because the chance of a false positive rises with the number of tests.

The authors wish to rule out mollusks as major contributors to sole diets based on the stable isotope analysis. Unfortunately, there wasn't a description what mollusks, if any, they used to develop the baseline isotope ratio for this group, and I didn't see them included in the stable isotope analysis. Also, while N and C are good, standard metrics, mixing models that use just these two and a subsample of baseline taxa may not always give accurate results. The authors need to add the mollusk isotope information and be more cautious in their interpretation.

Most of the conclusions are appropriate and base on statistical outcomes. However, results for stomach fullness were not significant, but seem to be discussed as real patterns regardless. More careful phrasing is needed in that part.

The writing is pretty good, but English is a difficult language. So, there are a lot of minor corrections and grammatical suggestions, and I imagine that the copy editor will have more. Having said this, the ms is perfectly understandable.

The figure and table legends are not 'stand alone' that is, many don't contain quite enough information to understand them without going back to the text. I've made several suggestions in this regard.

I elaborate on these comments in the attached pdf. The review program doesn't allow me to attach two files. My comment on the supplemental material is to make the legends 'stand alone' by listing the purpose of the study, species, site, and year. Also, this is where a lot of the apparently uncorrected T tests are given (needing Bonferroni correction or similar).

Reviewer 3 Report

The authors assessed the trophic ecology of the common sole Solea solea in the Adriatic sub-basin. The study provides new insights into the dietary resources for the species following a double approach: stomach content and isotopic analysis. The study involved a large set of specimens, and the results achieved are interesting. I enjoyed reading the manuscript. However, there are several important issues and considerations requiring proper addressing. Anyway, it is a well-done study that will deserve publication after a major review.

My main comments on the manuscript are the following:

Title

The reference to ontogenic diet shift should be removed since the authors did not study ontogenic changes but compared two size-class groups.

Simple summary

The content is too generalist. It should focus more on the study and the results achieved.

Abstract

Needs rewriting. The authors must provide a global but clear synthesis of the results achieved and their novelty and importance.

Introduction

This part is clear and well-written but it needs attention. This part is lacking in justification of the current study and approach and I would suggest the authors rethink how they are presenting their study. Also, the authors should provide more biological and ecological info on the target species.

On the other hand, it is unclear to me whether this is the first study on common sole following an isotopic approach. Otherwise, what is known?

L. 54: Review. Should read “… 2021). This ….”.

L. 72-73: Rephrase. Maybe: “….of both stomach contents (SCA) and signatures of stable isotopes (SIA) δ15N and δ13C.”

L. 75: Please, provide reviews for this topic instead of self-citations.

L. 87: There is a lot of self-citation in this paper. I recognize the authors need to refer to past work to justify this study, but there is more literature out there as well that is not presented. Some are unnecessary. Those on this line are an example. Please, reduce the number of self-citations and provide only those significant ones.

Materials and methods

L. 132-133: These self-citations are unnecessary.

L. 134: Please, confirm whether %GSI was calculated exclusively in females.

L. 150: H′ index is well known but provide a reference.

L. 165: Should read “… than 3 (Logan et al., 2008),”.

L. 166: Should read “… according to Post et al. (2007):….”.

L. 195: HSI and GSI were solely determined in females, that’s ok, but why didn’t you run GLM in males considering total length and weight as independent variables?

L. 213: Reference for Tilley et al., 2013 is lacking in the reference list. This reference refers to an isotopic study in an elasmobranch, the stingray Dasyatis americana. The authors must clearly support the use of TEFs from an elasmobranch, even if it has been used by the authors in another Mediterranean benthic mesopredator.

L. 215: Provide the full name for R. asterias.

L. 218-219: Delete of in “and of ichthyoplankton”.

L. 233-234: Use “Fanelli et al., unpublished data” instead of “…values from Fanelli E., personal communication)”.

Topic 2.2 Samples collection. The position of the sampling hauls seems to be affected by the seasonal period (see Figure 1). The reason underlying this sampling procedure should be explained. Is it based on migratory events?

Topic 2.4.2: About Permanova, please, indicate the package used in R (Adonis?). Also, provide info on nesting if applied.

Topic 2.4.2: Season and size were selected as factors in Permanova. Why not sex?

Topic 2.4.2: Why didn’t you run SIBER to assess niche widths and trophic preferences in males and females?  It would be very interesting to know about the differences between both sexes.

Results

L. 263-264: In females only?

L. 278: Delete “observed”.

L. 280: Delete “varied”.

L. 322: Delete “the” in “differences for both the factors”.

L. 333: Can “fish scales” be considered a taxon or traces resulting from fish digestion?

L. 352-353: It is stated that δ13C values or δ15N values were not correlated with TL. Was the analysis performed over the whole set of data? What if the analysis were done separately for each season and/or sex?

L. 357: I suggest “for factors “Season” and “Size”.

L. 359: Provide some data on C:N ratios.

L. 366-358:  It is stated in L. 356-359 that “PERMANOVA main test showed significant differences (p<0.05) in δ13C-δ15N values for both the factors “Season” “Size” (Suppl. Table 9). δ15N values resulted significantly different for the factor “Season” while those of δ13C varied significantly between the two levels of the factor “Size”. So, there were differences in δ13C  values due to size and season, and that finding is important when running Bayesian mixing models. However, the paragraph states the following: “Since no significant differences were found in δ15N values between the two class sizes, Bayesian mixing models were run on all the isotopic results without distinguishing between medium- and large-size specimens ”. Therefore, I do not understand why the Bayesian analysis was not run considering season and size, groups. This must be clearly explained.

L- 372: Use” proportional contribution”.

L. 390-392: Providing some data on niche overlap across seasons and/or sexes would enhance the manuscript.  I suggest the use of NicheRover v.1.1.0 in R.

Discussion

L- 406-408: Too many references.

L. 427: I suggest “also contains” instead of “is integrated with”.

L. 457: The authors claim that SIA results were consistent with SCA results. Isotopic values in muscle tissue usually reflect the diet ingested previously depending on the turnover rates of the target species. Generally, it implies that the diet would be reflected isotopically a few months later. The authors should discuss this subject. It would likely explain the biases between SCA and SIA results as pointed out in L. 465. It will likely imply a partial update and rewriting of the discussion.

L. 501-505: This part was not considered in the results and it has a secondary interest.

Conclusions

This part needs rewriting as it does not summarize efficiently the main results achieved.

It is stated that “From the data obtained, it is possible to confirm a strong variability in the diet that was related to both prey availability and energy requirements for reproduction”. Those statements are merely hypotheses not directly demonstrated by the study. slightly

L. 522: Delete “are”.

L. 523-524: The sentence “This paper provided important information for the management of this important resource for the Adriatic basin and the Mediterranean in general” is too vague. How it would contribute to that?

The discussion must address the subject referring to niche width considering seasonal changes (it is slightly discussed) and especially sexes.

Figure 2 Legend: Revise “medium-a”. Also, indicate figures A and B.

Figure 6: The horizontal line within the box represents the medians and the 95 % credibility intervals.

Supplementary Table 6:  Review “and seasonal differences diets of”.

Supplementary Table 7: Delete “(“ in “((H’)”.

Supplementary Table 10: Use “contribution” instead of “contribute”.

Supplementary Figure 2: The average isotopic values in fish larvae and the sea cucumber Paraleptopentacta elongate were almost the same, but both differ in ecological and trophic characteristics. How did you discriminate both (see sup. Table 10 and L. 382 in the main text)?

Supplementary Figure 2: Although the d13C and d15N data were provided in this figure, I would strongly recommend the authors use this plot as one of the main figures. Having this plot in the main text allows readers to check whether most of the S. solea d13C and d15N data fall inside the mixing polygon of the sources, which is the most fundamental assumption of the stable isotope mixing model. Similarly, I suggest including Supp. Figure 3 in the main manuscript.

After reading the manuscript, I understood that the assessment of the trophic position in S. solea across seasons (and sexes) was not among the aims of the study. However, providing that information would contribute to the strengthening of the study.

Round 2

Reviewer 3 Report

Dear authors, 

The new version of the manuscript has largely improved the first version. Most of my previous comments and suggestions have been properly adressed, and some inconsitencies are clear now. You have done a good job and the manuscript can be accepted in present form. Congratulations!

Regards